# Differentiating between Seronegative Elderly-Onset Rheumatoid Arthritis and Polymyalgia Rheumatica: A Qualitative Synthesis of Narrative Reviews

**DOI:** 10.3390/ijerph20031789

**Published:** 2023-01-18

**Authors:** Ryuichi Ohta, Chiaki Sano

**Affiliations:** 1Community Care, Unnan City Hospital, Unnan 699-1221, Japan; 2Department of Community Medicine Management, Faculty of Medicine, Shimane University, Izumo 690-0823, Japan

**Keywords:** seronegative, elderly, aged, older, rheumatoid arthritis, polymyalgia rheumatica, follow, differentiation

## Abstract

Elderly-onset rheumatoid arthritis (EORA) is prevalent among older patients, and its incidence is increasing due to aging societies. However, differentiating between EORA and polymyalgia rheumatica (PMR) is challenging for clinicians and hinders the initiation of effective treatment for rheumatoid arthritis among older generations, thereby allowing its progression. Therefore, we conducted a qualitative synthesis of narrative reviews via meta-ethnography regarding seronegative EORA diagnosis to clarify the methods to differentiate seronegative EORA from PMR. Three databases (PubMed, EMBASE, and Web of Science) were searched for relevant reviews published between January 2011 and October 2022. The extracted articles were synthesized using meta-ethnography, and 185 studies were selected following the protocol. Seven reviews were analyzed, and four themes and nine concepts were identified. The four themes included difficulty in differentiation, mandatory follow-up, and factors favoring rheumatoid arthritis and those favoring PMR. Factors favoring seronegative EORA and PMR should be considered for effective diagnosis and prompt initiation of disease-modifying anti-rheumatic drugs. Mandatory and long follow-ups of suspected patients are essential for differentiating the two diseases. The attitude of rheumatologists toward tentatively diagnosing seronegative EORA and flexibly modifying their hypotheses based on new or altered symptoms can aid in effective management and avoiding misdiagnosis.

## 1. Introduction

Early diagnosis and treatment of rheumatoid arthritis is essential to improving the quality of life of patients. Rheumatoid arthritis is prevalent in 0.5–1% of the total population, primarily occurs in middle-aged women [1], and commonly exists as peripheral arthritis with a progressive clinical course [2]. Rheumatoid arthritis is diagnosed based on classification criteria that comprise clinical findings, inflammatory markers, and autoantibodies [3]. Early treatment of rheumatoid arthritis is crucial to preventing progressive joint destruction [4,5]. Early initiation of disease-modifying anti-rheumatic drugs (DMARDs) can mitigate disease progression and preserve the quality of life of patients [4,5].

As society ages, the number of older patients suffering from rheumatoid arthritis increases [6]. Elderly-onset rheumatoid arthritis (EORA) is prevalent among older patients and differs in presentation from rheumatoid arthritis that affects younger generations. Additionally, rheumatoid arthritis in patients aged over 65 years has a different presentation than in younger patients [7]. The onset of EORA is more drastic than young-onset rheumatoid arthritis (YORA) [8]. Joint involvements may differ between EORA and YORA. EORA can involve proximal joints, such as the shoulder, neck, and femoral joints [9]. Furthermore, EORA and YORA vary genetically; namely, in the presence of HLA DRB1 [7]. To avoid misdiagnosing EORA, physicians should be aware of these varied clinical presentations.

The difficulty in differentiating between EORA and polymyalgia rheumatica (PMR) hinders the initiation of effective treatment for rheumatoid arthritis among older patients. PMR is another rheumatic disease that is common among older populations; its symptoms include musculoskeletal pains in the shoulders and girdles [10]. PMR symptoms can be similar to those of EORA when they appear peripherally [11]. Although joint swelling or evidence of synovitis/tenosynovitis is an important differentiating feature in EORA, these also might be an initial presentation in PMR due to high inflammation [10,11]. Furthermore, seronegative EORA can be difficult to differentiate from PMR because serological tests such as rheumatoid factors and anti-citrullinated protein antibodies are critical for the differential diagnosis [7]. In aging societies, differentiating seronegative EORA and PMR is a challenge for rheumatologists.

Clarifying the ways to differentiate between the two diseases can ensure effective initiation of DMARDs by clinicians to prevent the progression of rheumatoid arthritis. Several publications have reviewed the diagnosis and treatment of rheumatoid arthritis; however, only a few comprehensive reports described the differentiation between seronegative EORA and PMR [12,13,14]. Various reviews have partially dealt with this differentiation and suggested tips for it. To effectively diagnose EORA, it is important to clarify the qualitative summary of each tip. Seronegative EORA is a vague concept and is a challenge to diagnose for physicians. There are various qualitative descriptions regarding seronegative EORA in narrative reviews by experts in the field of rheumatology that include clinical tips. We consider the qualitative synthesis of descriptions of clinical tips based on their experiences and reviews to be beneficial in future studies of seronegative EORA. The qualitative synthesis of these descriptions is crucial to a comprehensive understanding of the differentiation between seronegative EORA and PMR. Therefore, our research question was: “How can clinicians differentiate seronegative EORA from PMR?” This study aimed to synthesize the differentiating factors between seronegative EORA and PMR among older patients through a meta-ethnography of narrative reviews of clinical tips.

## 2. Methods

We performed a qualitative synthesis using the meta-ethnography method, which is used for the synthesis of qualitative data [15,16]. The original articles on meta-ethnography suggested that this method can be also used for the synthesis of qualitative data in any scientific paper. Originally, meta-ethnography was developed to synthesize all qualitative studies that clarified the deep parts of the real world. In clinical medicine, there are various context-based experiences and knowledge of specialists summarized as narrative reviews. These experiences and wisdom of narrative reviews cannot be synthesized by using quantitative methods [15,16]. To synthesize these data, meta-ethnography can be a useful methodology. This process can be applied to the qualitative synthesis of narrative review articles [15,16]. We used meta-ethnography to synthesize qualitative evidence of narrative reviews regarding the differentiation between seronegative EORA and PMR.

Based on the research question, we decided on the search terms using the framework of population, types of study, and included contents. Our search terms were: “seronegative”, “elderly onset”, “late onset”, “rheumatoid arthritis”, “polymyalgia rheumatica”, and “review”. We searched for the relevant reviews on PubMed, Web of Science, and Embase to collect the related reviews comprehensively. The search strategy used was: “seronegative” and [“elderly onset” OR “late onset”] AND [rheumatoid arthritis] AND “polymyalgia rheumatica” AND “review.”

### 2.1. Study Selection

The inclusion and exclusion criteria are listed in Table 1. Narrative articles were included in the meta-analysis, whereas conference presentations, original articles, and duplicate articles were excluded.

### 2.2. Data Extraction

Literature searches and data extraction were independently conducted by the first investigator (R.O.), who extracted information regarding the differentiation between seronegative EORA and PMR from each review using a purpose-designed data extraction form. The second investigator (C.S.) checked the extracted contents, which were synthesized using meta-ethnography.

For credibility, the extracted data were discussed among the investigators. Any discrepancies were resolved via discussion with them. Databases were searched for narrative reviews regarding the differentiation between seronegative EORA and PMR. Studies without clear descriptions of the aims, participants, or outcomes were excluded (Table 1). In case of difficulty in the categorization and extraction of data, the investigators discussed the contents until agreement.

### 2.3. Statistical Analysis

A qualitative synthesis conducted via meta-ethnography was performed using the following eight steps: getting started, deciding what was relevant to our initial interest, reading the studies, determining how the studies were related, rereading the studies, translating the studies into one another, synthesizing translations, and detailing the synthesis [15,16]. The first step involved searching for related articles using search engines. To decide what was relevant to our initial interest, the first investigator selected reviews to be included in the meta-ethnography by reading the abstracts and checking for concordance with the inclusion criteria. Subsequently, the first investigator repeatedly read all the selected reviews and extracted the sections relevant to the differentiation between seronegative EORA and PMR. Vague sections were discussed with the second investigator to decide on their inclusion in the analysis. The studies were then translated into one another by inductively coding the extracted content. For the translation synthesis, we thematically synthesized the concepts and themes that appeared in each review. For triangulation, the concepts and themes were discussed among the researchers and also analyzed iteratively during the review period after the completion of a tentative analysis of reviews for theoretical saturation.

## 3. Results

Of the 185 studies analyzed, 5 were excluded due to duplication based on the names of authors and titles. After reviewing the abstracts, 159 studies were excluded because they did not include review content. Ultimately, 7 reviews were included in the final analysis after excluding 14 reviews based on the absence of contents regarding the differentiation between seronegative EORA and PMR (Figure 1). The seven articles are listed in Table 2.

### 3.1. Meta-Ethnography Results

Through meta-ethnography, four themes and nine concepts were identified (Table 3). The four themes included difficulty in differentiation, mandatory follow-up, and factors favoring rheumatoid arthritis and PMR. Each theme was described after reviewing the quotes and concepts appearing in the included articles.

### 3.2. Difficulty in Differentiation

Differentiating between seronegative EORA and PMR presents various difficulties, primarily due to two disease traits: symptom similarity at initial presentation and the clinical picture of EORA with PMR phenotype.

#### 3.2.1. Symptom Similarity at Initial Presentation

The presentation of EORA and PMR may be similar at the time of initial presentation. EORA may present with proximal joint and muscular symptoms that are similar to those of PMR. One review stated the difficulty in distinguishing between patients with PMR and those with early-stage EORA and PMR-like presentation [17]. It also mentioned that PMR symptoms are the principal initial manifestation in 25% of patients with EORA, of whom 10% eventually develop characteristic rheumatoid arthritis features [17].

This difficulty is caused by the presence of arthralgia due to joint pain in PMR. The presence of joint pain does not rule out the possibility of PMR. Accordingly, another review revealed not only the similarity between the two at the initial presentation but also the continuity from PMR to seronegative EORA, which may hinder diagnosis [21]. Furthermore, shoulder pain and arthritis can appear in both diseases; therefore, shoulder symptoms are not used for their differentiation. One review demonstrated that 13–23% of patients with early EORA have an explosive onset of shoulder arthritis, which resembles PMR symptoms [18].

#### 3.2.2. Clinical Picture of EORA with PMR Phenotype

One of the EORA presentations may include PMR symptoms throughout its clinical course. Differentiation from PMR can be achieved in the presence of autoimmune antibodies. Some types of seronegative EORA have symptoms that mirror those of PMR. One review suggested that one of the forms of EORA presentation is a PMR-like pattern that involves proximal limb joints and more acute disease onset with lower RF positivity and erosive disease [20]. Moreover, EORA displays a large joint involvement with the possibility of small joint involvement as well. Several reviews have suggested that there are three distinct clinical patterns within EORA’s heterogeneity. A PMR-like and usually RF-negative form has an acute onset, non-eroding joints, and a good prognosis. A quarter of PMR patients may also have asymmetric non-erosive polyarthritis, which highlights the need for a good differential diagnosis [19]. Another review reported that >20% of patients with PMR were later diagnosed with rheumatoid arthritis, which lent to the theory that PMR and EORA are components of a single disease process [21]. Some types of seronegative EORA involve PMR presentation; therefore, the differentiation from PMR can be challenging because this disease is a part of the EORA clinical disease picture.

### 3.3. Mandatory Follow-Up

For an effective diagnosis, mandatory follow-up for a duration of months and years is essential.

#### 3.3.1. Duration of Months and Years

Similar initial presentation and the clinical picture of EORA with PMR symptoms demand symptomatic follow-up of patients suspected to have seronegative EORA. One review suggested that follow-up is required in cases in which differentiating between PMR and seronegative EORA with PMR-like presentation is a challenge [18]. Moreover, the same review stated that patients who initially present with polymyalgia may evolve into a clinical situation closer to seronegative rheumatoid arthritis [18]. However, the duration of follow-up remains unclear. One review suggested, “A follow-up of several months may be required to make a definite distinction between PMR and EORA” [17]. In another, “EORA pattern is characterized by clinical and prognostic similarity to PMR. It is characterized by sudden onset, wrist tenosynovitis, common pitting edema in the hands, and spontaneous remission within 3–18 months” [19]. Another review reported, “The least common pattern is characterized by a sudden onset of symptoms, wrist tenosynovitis, pitting edema in the hands, and spontaneous remission within 3–18 months. Every attempt must be made to rule out other differential diagnoses (such as PMR, polyarticular gout, systemic vasculitis, and paraneoplastic manifestations) in older patients presenting with joint symptoms” [20].

Thus, year-long observations can help differentiate seronegative EORA from PMR because PMR can undergo remission in this duration.

#### 3.3.2. Peripheral Lesions Follow-Up Leading to EORA Diagnosis

During follow-up, peripheral joint pain and arthritis appear in patients with seronegative EORA. Therefore, a follow-up to detect peripheral lesions is useful for diagnosing seronegative EORA. One review suggested that approximately 50% of patients with PMR with peripheral lesions were diagnosed with EORA 1 year post-treatment [22]. The observation of patients suspected of having seronegative EORA can focus on the appearance of peripheral lesions to effectively diagnose the disease.

### 3.4. Factors Favoring Rheumatoid Arthritis

In the differentiation between seronegative EORA and PMR, initial radiographic change and the presence of peripheral arthritis are factors that favor seronegative EORA.

#### 3.4.1. Initial Radiographic Change

Rheumatoid arthritis is an inflammatory disease of the joints that progressively destroys them. Signs of deformities and joint destruction on radiography indicate the diagnosis of seronegative rheumatoid arthritis. One review suggested that “ACPA-positive EORA or ACPA-seronegative EORA with bone erosion at baseline is clearly different from PMR in terms of the progression of joint destruction, while seronegative EORA without bone erosion might have a benign course in terms of radiological joint destruction. Progression of bone erosion is essential for precise differential diagnosis, but rheumatoid factor- or anti-citrullinated protein antibody-negative early EORA with a PMR phenotype would not progress to erosive arthritis if initially treated with csDMARDs with or without GC therapy. These findings suggest a phenotypic overlap of PMR and seronegative early EORA with a PMR phenotype” [22]. To diagnose seronegative EORA at initial presentation, performing radiography is crucial for detecting the presence of deformities and erosions.

#### 3.4.2. Peripheral Arthritis

The presence of peripheral arthritis may indicate seronegative rheumatoid arthritis and differentiate it from PMR at initial presentation. Arthritis of various peripheral joints suggests seronegative EORA. One review suggested, “An erosive arthritis or the symmetrical involvement of metacarpophalangeal and/or proximal interphalangeal joints can help to diagnose seronegative EORA” [21]. Another reported, “The presence of metacarpophalangeal (MCP)/proximal interphalangeal (PIP) joint arthritis with proximal limb joint involvement is considered a predictive factor for seronegative EORA” [19]. Detecting peripheral arthritis such as MCP and PIP joint arthritis can aid in the effective diagnosis of seronegative EORA.

### 3.5. Factors Favoring Polymyalgia Rheumatica

In the differentiation between seronegative EORA and PMR, hip joint pain, extracapsular inflammation of peripheral lesions, and asymmetrical small joint involvement are factors that favor polymyalgia rheumatica.

#### 3.5.1. Hip Joint Pain

To diagnose seronegative EORA, the characteristics of PMR should be considered. Compared to EORA, PMR tends to involve the proximal joints with muscle pain with morning stiffness. One review suggested that in 50% and 20% of patients with PMR and EORA with shoulder lesions, respectively, pain and limited range of motion of the hip joints were reported; therefore, both shoulder and hip lesions are PMR phenotypic features [22]. The dominant symptoms such as hip and shoulder pain and limited range of motion of the joints indicate the possibility of PMR.

#### 3.5.2. Extracapsular Inflammation of Peripheral Lesions

In addition to joint and muscle pain, surrounding tissues are involved in PMR pathophysiology. Even if peripheral pain occurs, whether the pain originates from the joints should be examined. As one review suggested, “Extracapsular inflammation of peripheral lesions with tenosynovitis and surrounding pitting edema is reportedly characteristic of PMR” [22]. Examining the joints and extracapsular tissues is essential for the effective diagnosis of seronegative EORA. PMR should be suspected in the presence of extracapsular tissue inflammation.

#### 3.5.3. Asymmetrical Small Joint Involvement

The distribution of joint pain should be considered to differentiate seronegative EORA from PMR. When patients have peripheral symptoms, the laterality of the symptoms should be taken into account. One review showed that approximately 25% of patients with PMR present with peripheral synovitis that is frequently asymmetrical and non-erosive [8]. The peripheral joint examination should include an observation of the distribution of positive physical findings. When laterality exists in the findings, PMR should be considered as a possible diagnosis.

## 4. Discussion

This meta-ethnography of narrative reviews regarding the differentiation between seronegative EORA and PMR clarified four themes: difficulty in differentiation, mandatory follow-up, and factors favoring rheumatoid arthritis and those favoring PMR. Although the clear differentiation of the two diseases is challenging, factors favoring seronegative EORA and PMR should be considered for an effective diagnosis. The fundamental method for differentiating between the two diseases is the mandatory follow-up of suspected patients for months and years. The attitude of rheumatologists toward tentatively diagnosing seronegative EORA and flexibly changing their hypothesis based on the appearance or new or changed symptoms can be critical.

Differential diagnosis of seronegative EORA and PMR is difficult; therefore, a definitive diagnosis at the initial stage may be challenging. As this article showed, this difficulty mainly stems from symptom similarity at initial presentation and the clinical picture of EORA with a PMR phenotype. Therefore, clinicians may not be able to differentiate the two diseases in the initial phases [23,24]. Both diseases can have an acute onset with systemic symptoms rather than only joint pain in seropositive rheumatoid arthritis or YORA [25]. In addition, seronegative EORA can present with symptoms of PMR [23,24,26]. Thus, rheumatologists and family physicians who deal with these diseases should be careful when proposing a definitive diagnosis to suspected patients. For effective management, meticulous follow-up of symptoms is required.

Continuous follow-up can help effectively diagnose seronegative EORA because symptoms can change during the clinical course and reach the spectrum of either seronegative EORA or PMR. This study showed that mandatory follow-up involves durations of months and years in addition to follow-up of peripheral lesions for EORA diagnosis. The duration of follow-up varied depending on the review [8,17,18,19,20]. The clinical courses of seronegative EORA and PMR are acute and affect the activities of daily living in older patients [27,28]. Steroids can be initiated to mitigate the acute symptoms of seronegative EORA and PMR after ruling out other diseases such as bacteremia, sepsis, and other systemic inflammations [29]. While tapering steroids, several symptoms such as joint and muscular pain may appear [21]. PMR symptoms may not reappear for months and years after tapering steroids [17,21,30]; however, those of seronegative EORA may recur [26,31], which can be a sign of seronegative rheumatoid arthritis [26,31]. Furthermore, the appearance of peripheral arthritis during the follow-up period can be used to diagnose seronegative EORA. As this study showed, peripheral arthritis can appear during follow-up with steroid tapering. Thus, for an effective diagnosis, extensive follow-up is essential during steroid tapering, during which detecting recurrences and the appearance of peripheral arthritis is useful.

Investigating the factors that favor rheumatoid arthritis may aid in the diagnosis of seronegative rheumatoid arthritis after ruling out PMR and contribute to the smooth initiation of DMARDs for rheumatoid arthritis. The factors that favor rheumatoid arthritis are initial radiographic changes and the presence of peripheral arthritis. Radiographic changes in peripheral joints such as the PIP and MP joints suggest the possibility of seronegative EORA rather than PMR [32,33]. During the initial diagnosis of patients suspected to have seronegative EORA, peripheral radiography is essential for an effective diagnosis. In addition, initial examinations should focus on detecting peripheral arthritis to differentiate EORA and PMR because the presence of peripheral arthritis can indicate the possibility of seronegative EORA [31]. Effective and comprehensive physical joint examination is crucial to diagnose seronegative EORA. Regarding the treatment of seronegative EORA, prompt initiation of DMARDs is necessary to prevent joint destruction and deformities [4,5]. Based on these results, DMARDs should be prescribed for older patients suspected to have EORA with initial radiographic changes of the peripheral joints or physical findings of peripheral arthritis.

Contrary to the factors that favor seronegative EORA, PMR-favoring factors such as hip joint pain, extracapsular inflammation of peripheral lesions, and asymmetrical small joint involvement should be considered when predicting the clinical course. PMR can be a self-limiting condition in some cases; therefore, discontinuation of steroids without the use of DMARDs may be possible after steroid intake for a period ranging from months to 2 years [17,30]. Patients with hip pain, extracapsular inflammation, and asymmetrical small joint involvement should be followed up after tapering steroids without prescribing DMARDs.

This study had some limitations. Only a few review articles were found that investigated the differentiation between seronegative EORA and PMR. Rheumatology is more commonly studied among young to middle-aged patients; therefore, EORA and PMR may be differentiated according to the different clinical courses among older patients, but the research available is limited. Future studies should prospectively investigate ways to differentiate between these two diseases. In addition, this meta-ethnography did not include original articles to confirm the evidence present in the review articles. The inclusion criterion used in this study may have excluded some grey articles by experienced practitioners. Furthermore, due to accessibility limitations, this review may have missed relevant articles published in languages other than English. To overcome this limitation, we employed global search engines. Future studies can define the findings of this review as clinical factors and analyze older patients diagnosed with seronegative EORA and PMR to identify the sensitivity and specificity of each factor.

## 5. Conclusions

This meta-ethnography of narrative reviews regarding the differentiation between seronegative EORA and PMR clarified four themes: difficulty in differentiation, mandatory follow-up, and factors favoring rheumatoid arthritis and those favoring PMR. The favoring factors for both conditions should be checked for an effective diagnosis and a prompt initiation of DMARDs. Additionally, mandatory follow-up of suspected patients for months and years is essential for differentiating between the two diseases. The attitude of rheumatologists toward tentatively diagnosing seronegative EORA and flexibly changing their hypothesis based on the appearance of new or altered symptoms can be important. With the rise of aging societies globally, differentiating between EORA and PMR is becoming increasingly important for physicians.

## Figures and Tables

**Figure 1 ijerph-20-01789-f001:**
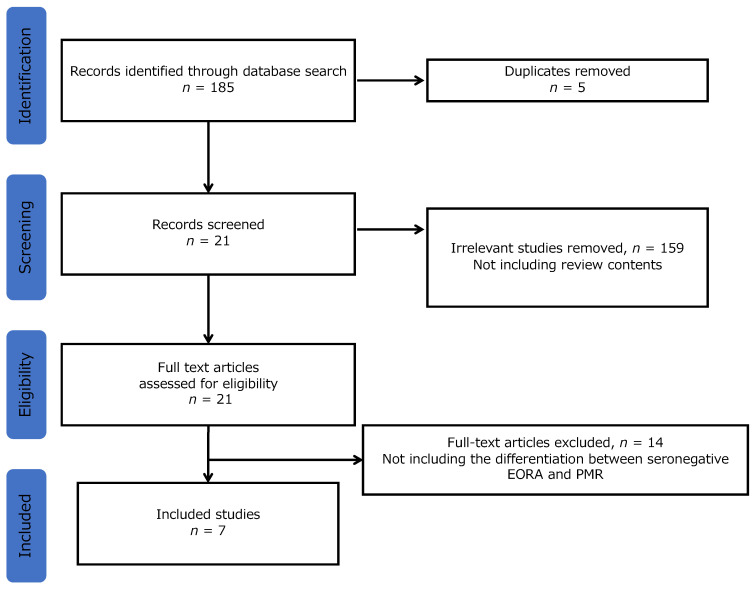
Selection flow. Note: EORA, elderly-onset rheumatoid arthritis; PMR, polymyalgia rheumatica.

**Table 1 ijerph-20-01789-t001:** Inclusion and exclusion criteria.

	Inclusion Criteria	Exclusion Criteria
Population	EORA	Other diseases
Types of study	Reviews	Original articles, non-empirical studies (editorial, news, and conference papers)
Included content	Differentiation between EORA and PMR	-
Other	Abstract available Full text available in English	Abstract not available Full text not available in English

Note: EORA, elderly-onset rheumatoid arthritis; PMR, polymyalgia rheumatica.

**Table 2 ijerph-20-01789-t002:** Studies included in the review.

Year	Title	Purpose	Included Codes
2009 [8]	Elderly Onset Rheumatoid Arthritis Differential Diagnosis and Choice of First Line and Subsequent Therapy	To review the EORA subset of patients with regard to demographic and clinical features, therapeutic options, outcomes, and differential diagnosis of EORA from other elderly rheumatological conditions	Duration (in months and years) Asymmetrical small joint involvement
2014 [17]	Polymyalgia rheumatica—diagnosis and classification	To differentiate PMR from other diseases	Symptomatic similarity at initial presentation Duration (in months and years)
2016 [18]	Targeting Low Disease Activity in Elderly-Onset Rheumatoid Arthritis: Current and Future Roles of Biological Disease-Modifying Antirheumatic Drugs	To review the clinical features of EORA and obstacles that prevent rheumatologists from providing standard treatment to patients with EORA	Symptom similarity at initial presentation Duration (in months and years)
2018 [19]	An autumn tale: geriatric rheumatoid arthritis	To review the clinical characteristics, prognosis, and treatment principles of EORA	Clinical picture of EORA with PMR phenotype Follow-up for peripheral lesions leading to EORA diagnosis Initial radiographic change Peripheral arthritis
2018 [20]	Morning Stiffness in Elderly Patients with Rheumatoid Arthritis: What is Known About the Effect of Biological and Targeted Agents?	To review the impact of morning stiffness in patients with rheumatoid arthritis and summarize the efficacy of the biologic and targeted synthetic disease-modifying anti-rheumatic drugs in the alleviation of morning stiffness	Clinical picture of EORA with PMR phenotype Duration (in months and years)
2019 [21]	Polymyalgia Rheumatica and Seronegative Elderly-Onset Rheumatoid Arthritis: Two Different Diseases with Many Similarities	To highlight the main differences and similarities between seronegative EORA, PMR, and PMR-like EORA	Symptom similarity at initial presentation Duration (in months and years) Initial radiographic change Peripheral arthritis
2022 [22]	Treatment strategies for elderly-onset rheumatoid arthritis in the new era	To review effective differential diagnosis and treatment for EORA	Peripheral pain occurring in EORA and PMR Hip joint pain favoring PMR Extracapsular inflammation of peripheral lesions suggesting PMR Follow-up of peripheral lesions leading to EORA diagnosis Initial radiographic change Peripheral arthritis

Note: EORA, elderly-onset rheumatoid arthritis; PMR, polymyalgia rheumatica.

**Table 3 ijerph-20-01789-t003:** Thematic analysis results in meta-ethnography.

Theme	Concept
Difficulty in differentiation	Symptom similarity at initial presentation
	Clinical picture of EORA with PMR phenotype
Mandatory follow-up	Duration (in months and years)
	Peripheral lesions follow-up leading to EORA diagnosis
Favoring rheumatoid arthritis	Initial radiographic change
	Peripheral arthritis
Favoring PMR	Hip joint pain
	Extracapsular inflammation of peripheral lesions
	Asymmetrical small joint involvement

EORA, elderly-onset rheumatoid arthritis; PMR, polymyalgia rheumatica.

## Data Availability

All relevant datasets used in this study are presented in the manuscript.

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
