# Peer review of "Differentiating between Seronegative Elderly-Onset Rheumatoid Arthritis and Polymyalgia Rheumatica: A Qualitative Synthesis of Narrative Reviews"

_ijerph, 2023, doi:10.3390/ijerph20031789_

Round 1

Reviewer 1 Report

I am very honored and grateful for the opportunity to peer review this very interesting work.

The authors conducted this study to clarify how to differentiate seronegative EORA from PMR, which are both ongoing problems in our aging society. The method chosen by the authors for this study was unique, as they sought to apply meta-ethnography, an integrative method of qualitative research that has been gradually gaining recognition in recent years.

The motivation for the authors to conduct this study is very valid, as the quick and reliable differentiation between EORA, which is progressive and should be treated with early medications, and PMR, which is basically self-limited, is very important for improving patient prognosis. Even though this is not a new original study, if new findings can be obtained through a comprehensive search and objective integration of previous findings, it would be a very valuable study.

On the other hand, I have doubts about the methodology chosen by the authors. The authors need to clearly argue the following points.

1. Is "qualitative integration of narrative reviews" appropriate as a study to find a point of differentiation between two diseases with distinctly different pathophysiology?

1-1. if we are to consider a method of disease differentiation by secondary research, a systematic review & meta-analysis of diagnostic accuracy, as has been done in many Cochrane reviews, for several noteworthy points (long-term follow-up, X-rays, peripheral arthritis, etc.) as pointed out by the authors would be the gold standard. Instead, is it appropriate to attempt "qualitative integration of narrative reviews"?

Narrative reviews are themselves highly biased works, and it is difficult to evaluate the quality and Risk of Bias of each review. The information obtained by integrating such narrative reviews, which contain bias that is difficult to evaluate, may not be comprehensive information.  Integration of narrative reviews would only produce one narrative review in the end.

1-3. can meta-ethnography integrate narrative reviews? The authors cited Sattar 2021, but meta-ethnography is not presented in Sattar 2021 as a method for integrating narrative reviews. It is only introduced as one method of integrating qualitative research. Meta-ethnography can integrate narrative studies (one of qualitative research), but it does not integrate narrative reviews. Returning to 1-1, in considering the differentiation of two different diseases, it should be based on quantitative studies, such as the combination and degree of symptoms, time course, and frequency, etc., not on qualitative studies.

1-4. It would be helpful to cite any previously published literature that analyzes the differentiation of diseases and diagnostic accuracy using the same methods as the authors (meta-ethnography of narrative reviews).

2. Assuming that the authors are able to provide a clear argument for point 1, then the validity of the search method is also questionable. The search strategy presented by the authors is very limited and is likely to miss many useful papers. Although the limitations of the search strategy are discussed to some extent in the Discussion, rather a systematic and comprehensive search should be conducted by using regular expressions such as *, OR instead of AND, and so on.

Author Response

Reviewer 1

I am very honored and grateful for the opportunity to peer review this very interesting work. The authors conducted this study to clarify how to differentiate seronegative EORA from PMR, which are both ongoing problems in our aging society. The method chosen by the authors for this study was unique, as they sought to apply meta-ethnography, an integrative method of qualitative research that has been gradually gaining recognition in recent years. The motivation for the authors to conduct this study is very valid, as the quick and reliable differentiation between EORA, which is progressive and should be treated with early medications, and PMR, which is basically self-limited, is very important for improving patient prognosis. Even though this is not a new original study, if new findings can be obtained through a comprehensive search and objective integration of previous findings, it would be a very valuable study.

  1. On the other hand, I have doubts about the methodology chosen by the authors. The authors need to clearly argue the following points.

 Is "qualitative integration of narrative reviews" appropriate as a study to find a point of differentiation between two diseases with distinctly different pathophysiology?

Response:

Thank you for the comment. We have revised the appropriate sentences to clarify our meaning. As the reviewer mentioned, meta-ethnography can be used for the synthesis of qualitative data. The original articles of meta-ethnography suggested that this method can also be used for the synthesis of qualitative data in any scientific papers. Originally, meta-ethnography was developed to synthesize all qualitative studies which clarified the deep parts of the real world. In clinical medicine, there are various context-based experiences and knowledge of specialists. These experiences and wisdom cannot be synthesized by using quantitative methods. To synthesize these data, meta-ethnography can be one of the useful methodologies. Therefore, we have changed the expression to qualitative synthesis respecting meta-ethnography.

  1. if we are to consider a method of disease differentiation by secondary research, a systematic review & meta-analysis of diagnostic accuracy, as has been done in many Cochrane reviews, for several noteworthy points (long-term follow-up, X-rays, peripheral arthritis, etc.) as pointed out by the authors would be the gold standard. Instead, is it appropriate to attempt "qualitative integration of narrative reviews"?

Response:

Thank you for the useful feedback. We agree that clarifying the qualitative summary of each topic is essential to effectively diagnose EORA. As we described in the Introduction, seronegative EORA is a vague concept and can be a challenge to diagnose for physicians. There are various qualitative descriptions and clinical tips regarding seronegative EORA mentioned in the narrative reviews by experts from the field of rheumatology. We considered the qualitative synthesis of descriptions of clinical tips based on their experiences and reviews are beneficial for future studies of seronegative EORA.

  1. Narrative reviews are themselves highly biased works, and it is difficult to evaluate the quality and Risk of Bias of each review. The information obtained by integrating such narrative reviews, which contain bias that is difficult to evaluate, may not be comprehensive information. Integration of narrative reviews would only produce one narrative review in the end.

Response:

Thank you for the useful feedback. We agree with the suggestion. For improving the credibility of this study, we have added the description of qualitative study in the Methods sections as follows:

Page 3, lines 9 – 38:

2.2 Data extraction

“Literature searches and data extraction were independently conducted by the first investigator (RO) who extracted information regarding the differentiation between seronegative EORA and PMR from each review using a purpose-designed data extraction form. The second investigator (CS) checked the extracted contents, which were synthesized using meta-ethnography.

For credibility, the extracted data were discussed among the investigators. Any discrepancies were resolved through discussion with them. Databases were searched for narrative reviews regarding the differentiation between seronegative EORA and PMR. Studies without clear descriptions of the aims, participants, or outcomes were excluded (Table 1). In case of difficulty in the categorization and extraction of data, the investigators discussed the contents until agreement.

2.3. Statistical analysis

Qualitative synthesis by meta-ethnography was performed using the following eight steps: getting started, deciding what is relevant to our initial interest, reading the studies, determining how the studies are related, rereading the studies, translating the studies into one another, synthesizing translations, and detailing the synthesis [15,16]. The first step involved searching for related articles using search engines. To decide what was relevant to our initial interest, the first investigator selected reviews to be included in the meta-ethnography by reading the abstracts and checking for concordance with the inclusion criteria. Subsequently, the first investigator repeatedly read all the selected reviews, extracting the sections relevant to the differentiation between seronegative EORA and PMR. Vague sections were discussed with the second investigator to decide on their inclusion in the analysis. The studies were then translated into one another by inductively coding the extracted content. For translation synthesis, we thematically synthesized the concepts and themes that appeared in each review. For triangulation, the concepts and themes were discussed among the researchers, and they were also analyzed iteratively during the review period after the completion of a tentative analysis of reviews for theoretical saturation.”

  1. Can meta-ethnography integrate narrative reviews? The authors cited Sattar 2021, but meta-ethnography is not presented in Sattar 2021 as a method for integrating narrative reviews. It is only introduced as one method of integrating qualitative research. Meta-ethnography can integrate narrative studies (one of qualitative research), but it does not integrate narrative reviews. Returning to 1-1, in considering the differentiation of two different diseases, it should be based on quantitative studies, such as the combination and degree of symptoms, time course, and frequency, etc., not on qualitative studies. It would be helpful to cite any previously published literature that analyzes the differentiation of diseases and diagnostic accuracy using the same methods as the authors (meta-ethnography of narrative reviews).

Response:

Thank you for the useful feedback. We agree with the suggestion and have revised our manuscript. As the reviewer suggested, meta-ethnography can be used for the synthesis of qualitative data. As the reviewer mentioned, the original articles of meta-ethnography suggested that meta-ethnography can be also used for the synthesis of qualitative data of any scientific papers.

Page 2, lines 37 – 43:

“Originally, meta-ethnography was developed to synthesize all qualitative studies which clarified the deep parts of the real world. In clinical medicine, there are various context-based experiences and knowledge of specialists. These experiences and wisdom cannot be synthesized by using quantitative methods. To synthesize these data, meta-ethnography can be a useful methodology.”

Besides, we have added several references to support our research methods in the Background and Methods sections.

  1. Assuming that the authors are able to provide a clear argument for point 1, then the validity of the search method is also questionable. The search strategy presented by the authors is very limited and is likely to miss many useful papers. Although the limitations of the search strategy are discussed to some extent in the Discussion, rather a systematic and comprehensive search should be conducted by using regular expressions such as *, OR instead of AND, and so on.

Response:

Thank you for the useful feedback. We agree with the suggestion. We needed to narrow down the search to answer the research questions specifically. We have described our processes more in-depth in the Methods section as follows:

Page 2, lines 46 – 52:

“Based on the research question, we decided on the search terms from the framework of population, types of study, and included contents. Our search terms were “seronegative,” “elderly onset,” “late onset,” “rheumatoid arthritis,” “polymyalgia rheumatica,” and “review”. We searched for the relevant reviews on PubMed, Web of Science, and Embase to collect related reviews comprehensively. The search strategies used were “seronegative” and [“elderly onset” OR “late onset”] AND [rheumatoid arthritis] AND “polymyalgia rheumatica” AND “review.”

Reviewer 2 Report

I have read your manuscript with interest. Although it addresses an important question of how to differentiate elderly-onset rheumatoid arthritis from polymyalgia rheumatica, the manuscript requires major revision.

1 Please use the Mesh words in the keywords.

2 Please move content starts with ‘’early diagnosis and treatment’’ from line 34 to 38 down and write them after finishing with epidemiology data.

3 Rephrase line 39 As society ages

4 These sentences start from ‘’elderly onset rheumatoid arthritis (EORA)’’ 40-44 lines sound repetitive and fragmented. You may rephrase and make two sentences.

5 ‘’Additionally, rheumatoid arthritis in patients aged over 65 years has a different presentation than in older patients [7]. EORA is more acute than young-onset rheumatoid arthritis (YORA)’’

This sentence is confusing and doesn’t make any sense. >65 is already old age. I suggest removing it and amend the manuscript accordingly.

6 Joint involvement can differ between EORA and YORA is not entirely true. EORA also affects small joints of hands and feet.  Does EORA affecting the proximal joints, such as the shoulder, neck, and femoral joints is due to osteoarthritis?

7 EORA doesn’t just have pain in the proximal joints. Joint swelling or evidence of synovitis/tenosynovitis is an important differentiating feature.

8 Please remove ‘care to older patients’ heading

9 Therefore, this study aimed to clarify realistic approaches to diagnose seronegative EORA differentially from PMR among older patients, through meta-ethnography of narrative reviews.

Please rewrite the sentence. It is not clear.

10 Please explain in the methods how you dealt with duplicates. And how did you do bias assessment?

11 Meta-ethnography is typically used for the synthesis of qualitative research [15]; various research documents can be qualitatively synthesized.

This sentence needs rewriting.

12 Please add a subtitle and describe more how you conducted the search.

13 In the Table 2, please add a column summarising each study’s findings. Almost all studies have the same design. So you can remove it.

14 ‘’One review suggested that one of the forms of EORA presentation is a PMR-like pattern, involving proximal limb joints and a more acute disease onset, with lower RF positivity and erosive disease [20]; these characteristics indicate a better prognosis.’’

This is confusing. How those characteristics indicate better prognosis. What indicates poor prognosis?

15 Can you compare and contrast with other previous studies that may have identified the same or different themes four themes (difficulty in differentiation, mandatory follow-up, and factors favoring rheumatoid arthritis and PMR) in this area and discuss the possible reasons?

16 Add future study recommendations solving your limitations. I would be better to add the clinical implications and some strengths too. 

Author Response

Reviewer 2

I have read your manuscript with interest. Although it addresses an important question of how to differentiate elderly-onset rheumatoid arthritis from polymyalgia rheumatica, the manuscript requires major revision.

  1. Please use the Mesh words in the keywords.

Response:

Thank you for the useful feedback. We agree with the suggestion and have revised the Keywords as per the Mesh words.

Page 1, lines 28 – 29:

Keywords: seronegative, elderly, aged, older, rheumatoid arthritis, polymyalgia rheumatica, follow, differentiation

  1. Please move content starts with ‘’early diagnosis and treatment’’ from line 34 to 38 down and write them after finishing with epidemiology data.

Response:

Thank you for the useful feedback. We agree with the suggestion and revised the Introduction comprehensively as follows.

Page 1, lines 30 – 38:

“Early diagnosis and treatment of rheumatoid arthritis is essential for improving the quality of life of patients. Rheumatoid arthritis is prevalent in 0.5–1% of the total population and primarily occurs in middle-aged women [1], commonly as peripheral arthritis with a progressive clinical course [2]. Rheumatoid arthritis is diagnosed based on classification criteria comprising clinical findings, inflammatory markers, and autoantibodies [3]. Early treatment of rheumatoid arthritis is crucial for preventing progressive joint destruction [4,5]. Early initiation of disease-modifying anti-rheumatic drugs (DMARDs) can mitigate disease progression and preserve the quality of life of patients [4,5].”

  1. Rephrase line 39 As society ages

Response:

Thank you for the useful feedback. We agree with the suggestion and have revised the sentence as follows.

Page 1, lines 39 – 42:

“As the society ages, the number of older patients suffering from rheumatoid arthritis increases [6]. Elderly onset rheumatoid arthritis (EORA) is prevalent among older patients and differs in presentation from rheumatoid arthritis affected younger generations.”

  1. These sentences start from ‘’elderly onset rheumatoid arthritis (EORA)’’ 40-44 lines sound repetitive and fragmented. You may rephrase and make two sentences. ‘’Additionally, rheumatoid arthritis in patients aged over 65 years has a different presentation than in older patients [7]. EORA is more acute than young-onset rheumatoid arthritis (YORA)’’ This sentence is confusing and doesn’t make any sense. >65 is already old age. I suggest removing it and amend the manuscript accordingly. Joint involvement can differ between EORA and YORA is not entirely true. EORA also affects small joints of hands and feet. Does EORA affecting the proximal joints, such as the shoulder, neck, and femoral joints is due to osteoarthritis?

Response:

Thank you for the useful feedback. We agree with the suggestion and revised the sentences repecting the reviewer’s comments including the difference of EORA and YORA as follows:

Page 1, lines 39 – 45; Page 2, lines 1 – 3:

“As the society ages, the number of older patients suffering from rheumatoid arthritis increases [6]. Elderly onset rheumatoid arthritis (EORA) is prevalent among older patients and differs in presentation from rheumatoid arthritis affecting younger generations. Additionally, rheumatoid arthritis in patients aged over 65 years has a different presentation than in older patients [7]. The onset of EORA is more drastic than young-onset rheumatoid arthritis (YORA) [8]. Joint involvements may differ between EORA and YORA. EORA can involve proximal joints, such as the shoulder, neck, and femoral joints [9]. Furthermore, EORA and YORA vary genetically, namely, in the presence of HLA DRB1 [7]. To avoid misdiagnosing EORA, physicians should be aware of these varied clinical presentations.”

  1. EORA doesn’t just have pain in the proximal joints. Joint swelling or evidence of synovitis/tenosynovitis is an important differentiating feature.

Response:

Thank you for the useful feedback. We agree with the suggestion and have revised the suggested parts accordingly as follows:

Page 2, lines 4 – 14:

“The difficulty in differentiating between EORA and polymyalgia rheumatica (PMR) hinders the initiation of effective treatment for rheumatoid arthritis among older patients. PMR is another rheumatic disease common among older populations, with symptoms of musculoskeletal pain in the shoulders and girdles [10]. PMR symptoms can be similar to those of EORA when they appear peripherally [11]. Although joint swelling or evidence of synovitis/tenosynovitis is an important differentiating feature in EORA, these might be an initial presentation in PMR also because of high inflammation [10,11]. Furthermore, seronegative EORA can be difficult to differentiate from PMR, because serological tests, such as rheumatoid factors and anti-citrullinated protein antibodies, are critical for the differential diagnosis [7]. In aging societies, differentiating seronegative EORA and PMR is a challenge for rheumatologists.”

  1. Please remove ‘care to older patients’ heading

Response:

Thank you for the useful feedback. We agree with the suggestion and have deleted the suggested phrase.

  1. Therefore, this study aimed to clarify realistic approaches to diagnose seronegative EORA differentially from PMR among older patients, through meta-ethnography of narrative reviews. Please rewrite the sentence. It is not clear.

Response:

Thank you for the useful feedback. We agree with the suggestion and have revised the sentences as follows:

Page 2, lines 20 – 31:

“To effectively diagnose EORA, it is important to clarify the qualitative summary of each tip. Seronegative EORA is a vague concept and is a challenge to diagnose for physicians. There are various qualitative descriptions regarding seronegative EORA in narrative reviews by experts in the field of rheumatology including clinical tips. We considered the qualitative synthesis of descriptions of clinical tips based on their experiences and reviews to be beneficial for future studies of seronegative EORA. The qualitative synthesis of these descriptions is crucial for a comprehensive understanding of the differentiation between seronegative EORA and PMR. Therefore, our research question is “How can clinicians differentiate seronegative EORA from PMR?” This study aimed to synthesize the differentiating factors between seronegative EORA and PMR among older patients, through meta-ethnography of narrative reviews of clinical tips.”

  1. Please explain in the methods how you dealt with duplicates. And how did you do bias assessment?

Response:

Thank you for the useful feedback. We have provided the following explanation in the Results section:

Page 4, lines 2 – 3:

“Of the 185 studies analyzed, five were excluded because of duplication based on the name of authors and titles.”

In additions, we have added the description of the quality improvement mitigating bias in the analysis sections as follows:

“Qualitative synthesis respected by meta-ethnography was performed using the following eight steps: getting started, deciding what is relevant to our initial interest, reading the studies, determining how the studies are related, rereading the studies, translating the studies into one another, synthesizing translations, and detailing the synthesis [15,16]. The first step involved searching for related articles using search engines. To decide what was relevant to our initial interest, the first investigator selected reviews to be included in the meta-ethnography by reading the abstracts and checking for concordance with the inclusion criteria. Subsequently, the first investigator repeatedly read all the selected reviews, extracting the sections relevant to the differentiation between seronegative EORA and PMR. Vague sections were discussed with the second investigator to decide on their inclusion in the analysis. The studies were then translated into one another by inductively coding the extracted content. For translation synthesis, we thematically synthesized the concepts and themes that appeared in each review. For triangulation, the concepts and themes were discussed among the researchers, and they were also analyzed iteratively during the review period after the completion of a tentative analysis of reviews for theoretical saturation.”

  1. Meta-ethnography is typically used for the synthesis of qualitative research [15]; various research documents can be qualitatively synthesized. This sentence needs rewriting.

Response:

Thank you for the useful feedback. We agree with the suggestion and have revised as follows:

Page 2, lines 34 – 45:

“We performed a qualitative synthesis using the meta-ethnography method which is used for the synthesis of qualitative data [15,16]. The original articles on meta-ethnography suggested that this method can be also used for the synthesis of qualitative data of any scientific papers. Originally, meta-ethnography was developed to synthesize all qualitative studies which clarified the deep parts of the real world. In clinical medicine, there are various context-based experiences and knowledge of specialists summarized as narrative reviews. These experiences and wisdom of narrative reviews cannot be synthesized by using quantitative methods [15,16]. To synthesize these data, meta-ethnography can be a useful methodology. This process can be applied to the qualitative synthesis of narrative review articles [15,16]. We used meta-ethnography to synthesize qualitative evidence of narrative reviews regarding the differentiation between seronegative EORA and PMR.”

  1. Please add a subtitle and describe more how you conducted the search.

Response:

Thank you for the useful feedback. We have revised the methodology to describe our process in detail as follows:

Page 3, lines 1 – 38:

2.1. Study selection

The inclusion and exclusion criteria are listed in Table 1. Narrative articles were included in the meta-analysis, whereas conference presentations, original articles, and duplicate articles were excluded.

Table 1. Inclusion and exclusion criteria

Inclusion Criteria

Exclusion Criteria

Population

EORA

Other diseases

Types of study

Reviews

Original articles, non-empirical studies (editorial, news, and conference papers)

Included content

Differentiation between EORA and PMR

-

Other

Abstract available

Full text available in English

Abstract not available

Full text not available in English

EORA, elderly onset rheumatoid arthritis; PMR, polymyalgia rheumatica

2.2. Data extraction

Literature searches and data extraction were independently conducted by the first investigator (RO) who extracted information regarding the differentiation between seronegative EORA and PMR from each review using a purpose-designed data extraction form. The second investigator (CS) checked the extracted contents, which were synthesized using meta-ethnography.

For credibility, the extracted data were discussed among the investigators. Any discrepancies were resolved through discussion with them. Databases were searched for narrative reviews regarding the differentiation between seronegative EORA and PMR. Studies without clear descriptions of the aims, participants, or outcomes were excluded (Table 1). In case of difficulty in the categorization and extraction of data, the investigators discussed the contents until agreement.

2.3. Statistical analysis

Qualitative synthesis by meta-ethnography was performed using the following eight steps: getting started, deciding what is relevant to our initial interest, reading the studies, determining how the studies are related, rereading the studies, translating the studies into one another, synthesizing translations, and detailing the synthesis [15,16]. The first step involved searching for related articles using search engines. To decide what was relevant to our initial interest, the first investigator selected reviews to be included in the meta-ethnography by reading the abstracts and checking for concordance with the inclusion criteria. Subsequently, the first investigator repeatedly read all the selected reviews, extracting the sections relevant to the differentiation between seronegative EORA and PMR. Vague sections were discussed with the second investigator to decide on their inclusion in the analysis. The studies were then translated into one another by inductively coding the extracted content. For translation synthesis, we thematically synthesized the concepts and themes that appeared in each review. For triangulation, the concepts and themes were discussed among the researchers, and they were also analyzed iteratively during the review period after completion of a tentative analysis of reviews for theoretical saturation.”

  1. In the Table 2, please add a column summarizing each study’s findings. Almost all studies have the same design. So you can remove it.

Response:

Thank you for the useful feedback. We agree with the suggestion and have deleted the study design. Besides, the study contents also contain similar expressions and findings except for the description of differentiation between EORA and PMR. Therefore, we did not add the study findings in Table 2.

  1. ‘’One review suggested that one of the forms of EORA presentation is a PMR-like pattern, involving proximal limb joints and a more acute disease onset, with lower RF positivity and erosive disease [20]; these characteristics indicate a better prognosis.‘‘ This is confusing. How those characteristics indicate better prognosis. What indicates poor prognosis?

Response:

Thank you for the useful feedback. We agree with the suggestion and have deleted this sentence because of its misleading nature.

  1. Can you compare and contrast with other previous studies that may have identified the same or different themes (four themes - difficulty in differentiation, mandatory follow-up, and factors favoring rheumatoid arthritis and PMR) in this area and discuss the possible reasons?

Response:

Thank you for the useful feedback. We agree with the suggestion and have revised the Discussion section comprehensively as follows:

Page 8, lines 47 – 50:

“Therefore, clinicians may not be able to differentiate the two diseases in the initial phases [23,24]. Both diseases can have an acute onset with systemic symptoms rather than only joint pain in seropositive rheumatoid arthritis or YORA [25]. In addition, seronegative EORA can present with symptoms of PMR [23,24,26].”

Page 9, lines 7 – 12:

“The duration of follow-up varies depending on the review [8,17–20]. The clinical courses of seronegative EORA and PMR are acute and affect the activities of daily living in older patients [27,28]. Steroids can be initiated to mitigate the acute symptoms of seronegative EORA and PMR after ruling out other diseases, such as bacteremia, sepsis, and other systemic inflammations [29]. While tapering steroids, several symptoms, such as joint and muscular pain, may appear [21].”

Page 9, lines 22 – 29:

“The factors favoring rheumatoid arthritis are initial radiographic changes and the presence of peripheral arthritis. Radiographic changes in peripheral joints, such as the PIP and MP joints, suggest the possibility of seronegative EORA rather than PMR [32,33]. During the initial diagnosis of patients suspected to have seronegative EORA, peripheral radiography is essential for effective diagnosis. In addition, initial examinations should focus on detecting peripheral arthritis to differentiate EORA and PMR as the presence of peripheral arthritis can indicate the possibility of seronegative EORA [31].”

Page 9, lines 38 – 42:

“PMR can be a self-limiting condition in some cases; therefore, discontinuation of steroids without the use of DMARDs may be possible after steroid intake for a period ranging from months to 2 years [17,30]. Patients with hip pain, extracapsular inflammation, and asymmetrical small joint involvement should be followed up after tapering steroids without prescribing DMARDs.”

  1. Add future study recommendations solving your limitations. It would be better to add the clinical implications and some strengths too.

Response:

Thank you for the useful feedback. We agree with the suggestion and have added the suggestion for future studies as follows:

Page 9, lines 49 – 55:

“The inclusion criterion used in this study may exclude some grey articles by experienced practitioners. Furthermore, owing to accessibility limitations, this review may have missed relevant articles published in languages other than English. To overcome this limitation, we employed global search engines. Future studies can define the findings of this review as clinical factors and analyze older patients diagnosed with seronegative EORA and PMR to identify the sensitivity and specificity of each factor.”

Round 2

Reviewer 1 Report

I would be grateful for the authors' appropriate response to my inquiry. I feel that the methodology and limitations are adequately mentioned and fairly reported.

Reviewer 2 Report

Thank you for revising the manuscript to incorporate the changes we proposed.